# The Impact of Pb from Ammunition on the Vegetation of a Bird Shooting Range

**Eva de la Peña [1,2,*], José Manuel Seoane [1] and Juan Carranza [1]**

[1] Wildlife Research Unit (UIRCP), University of Córdoba, 14071 Córdoba, Spain; jmseoaner@gmail.com (J.M.S.); jcarranza@uco.es (J.C.)

[2] Instituto de Investigación en Recursos Cinegéticos, IREC (CSIC-UCLM-JCCM), 13071 Ciudad Real, Spain

[*] Correspondence: evadelapenha@gmail.com; Tel.: +34-6903-93-915

**Abstract:** Hunting with lead ammunition represents a source of heavy metal pollution to the environment that can be potentially high at the local scale. Intensive hunting of small game species can concentrate high levels of ammunition discharging in small areas. This type of hunting is a relevant economic resource for private landowners in some regions of Spain, and current legislation allows the use of lead ammunition in these scenarios. It becomes, therefore, highly relevant to study whether this activity may pose concerns to the conservation of the environment in the areas where it takes place. Using a red-legged partridge (*Alectoris rufa*) shooting range as a study area, we examined the effect of intensive hunting on this species on the vegetation present. We found significantly higher lead levels in the sprouts of plants of shooting areas related to control sites of the same property where partridge shooting does not occur. We found differences in the presence of lead between sprouts of different plant species. In addition, old sprouts of existing vegetation in shooting areas also showed higher lead levels than newly emerged sprouts of the same plants. These results demonstrate the impact of lead ammunition on vegetation in terms of persistence over time and differences between species. Further analyses using chemical and ecotoxicological data are necessary to evaluate the extent of environmental pollution risks. Our results provide new support in favor of the use of alternative ammunition, with particular emphasis on scenarios where hunting activity is intensive.

**Keywords:** intensive hunting activity; red-legged partridge; ammunition; lead (Pb) levels; vegetation; sprouts; shooting area; non-shooting area

## 1. Introduction

Despite heavy metals, such as lead (Pb), which occur naturally on the earth's surface numerous previous studies suggest that the extensive release of Pb in nature has been largely the outcome of human activities. This includes mining [1–3], industrial waste [4], pollution in past years from lead in gasoline [5,6] and Pb ammunition still used for hunting and civilian/military shooting activities [7–10]. The necessity for information on the status of Pb in the environment is relevant when economically human activities are at stake. Besides, the accumulation of Pb concentrations in food plant species causes different magnitudes of risks to human health [11–13].

One of the most relevant impacts caused by the concentration of hunting activity in a specific area is the accumulation of Pb from pellets deposited in the topsoil [13]. Pb is a chemical element that, in the case of hunting ammunition, occurs in metallic form and alloys with other metals. Pb shot degrades into particulate or molecular Pb species [14] at a specific rate that depends on the environmental characteristics of the site (e.g., soil acidity, organic matter content, etc.). Depending on both soil use and characteristics [15] and agricultural practice (e.g., the application of sludge to agricultural land [16]), the persistence of Pb pellets in the soil varies between 30 and 300 years. Pb in the environment

accumulates in sediments but may also be present as charged or complexed ions in interstitial and estuarine waters or in particulate phases that drive its mobility [17]. Over time, accumulated Pb tends to penetrate deeper layers of both soil and water, resulting in heavy metal contamination [18]. Pb in hunting and shooting range areas also represents an environmental issue due to their potential movement through the topsoil to groundwater and surface water [19]. This fact implies that it can be ingested by birds or soil biotas such as earthworms [20] or transferred to crops [21].

Pb poisoning in wildlife, as well as in humans, is a widely studied and acknowledged problem [22,23]. Adverse effects had been already reported on public health [24], reproductive [23,25], behavioral [26,27] and, even a lethal threat for many endangered species (e.g., white-headed duck, *Oxyura leucocephala*; California Condor, *Gymnogyps californianus*). Apart from direct mortality, a wide range of sublethal effects has also been associated with Pb exposure in birds [28] regarding immune function [29–31]. Previous studies have shown the consequences of Pb exposure in immune function and the different ways in which individuals can counteract these effects among seasons [23,31].

The highest densities of Pb accumulation occur where intensive hunting activity is carried out from fixed hunting positions and over long-term periods [32]. Intensive hunting involves the concentration in space and time of ammunition released into the environment deposited mainly on the soil. Previous studies highlighted several adverse effects of intensive hunting as avian plumbism and implications for game meat security in wetland areas across Europe [32]. Furthermore, there is previous evidence of Pb contamination in topsoil due to hunting activities in different places around the world (i.e., dove hunting in Finland [33] and in the Czech Republic [20]).

Legislation forbids hunting with Pb ammunition in wetlands (The Natural Heritage and Biodiversity Law (33/2015) due to the impact of Pb accumulation in these habitats [29]. The use of Pb ammunition results in the deposition of considerable quantities of this heavy metal in the environment [34], especially in surface soil layers and at the depths of watercourses where it is accumulated by runoff [35]. In plants, the most Pb accumulation is in roots because of the contaminated soil adhering to this part of plants, but Pb can move from roots to plant tissues [36]. Pb is often highly mobile and highly photo available due to their weathering, so its uptake by plants can be a risk or an advantage when considering its role in phytoremediation (e.g., *Bituminaria bituminosa* and *Atriplex halimus* [37–39]).

The adverse effects of Pb in plants include disorders that upset normal physiological activities of the plants [40] regarding the inhibition of photosynthesis [41], germination of seeds and growth development [42], even cell death [43]. Specifically, Pb accumulation impairs the action of the relevancy determinant enzyme in several essential plant metabolic pathways, involved in both nitrogen metabolism [44] and sugar metabolism [45] but also the Calvin cycle [46]. Concerning this, leaves are the most relevant organs in plants due to their role in light uptake during photosynthesis [40,47]. Pb is absorbed very rapidly once it gains access to internal tissues and moves through vascular tissues and accumulates in the distal parts of the plant [48], being key bio-indicators in the accumulation of Pb released into the environment.

In Spain, intensive hunting is restricted to specific sites. The main purpose is hunting using periodic releases of small game reared on certified hunting farms, where appropriate, or in which small game species are regularly restocked. This context requires the analysis of the interactions between the management model and the effects on the environmental values surrounding these areas. The setting of a management protocol is needed to ensure compatibility between an economic benefit and the maintenance of the environment and biodiversity conservation. One of the most relevant impacts in these scenarios is the discharge of elevated ammunition due to the concentration of intensive hunting activity in time and space. Thus, investigating the short- to medium-term effects on the environment (e.g., vegetation) is key to monitoring and evaluating its impact.

Our main goal was to determine accumulated Pb in an intensive hunting estate by analyzing vegetation accumulated from shooting selected sites to (1) detect the influence of hunting on Pb accumulated variations, (2) explore the differences between species plants in the shooting influenced area, and (3) approximate the potential environmental threat of anthropogenic Pb for intensive hunting management (in terms of total concentrations present in vegetation).

## 2. Materials and Methods

### 2.1. Study Site

The study was carried out within a hunting property with a surface area of 1343 ha located in the Sierra Morena mountains in the South of Spain (Figure 1). The perimeter of the estate is 21,220 m. It borders to the north with the Sierra Norte de Sevilla Natural Park (Andalucía, Spain), which is one of the protected natural areas that make up the European ecological network "Natura 2000", established by Directive 92/43/EEC (Habitats Directive), being a Special Area of Conservation ('ZEC'), and a Special Protection Area for Birds ('ZEPA') established under the Birds Directive. On the northern boundary of the property, there are two sites catalogued as Sites of Community Interest ('LIC'). These protected areas protect these ecosystems to contribute to guaranteeing the conservation of biodiversity by preserving the natural habitats and wild fauna and flora of the territory.

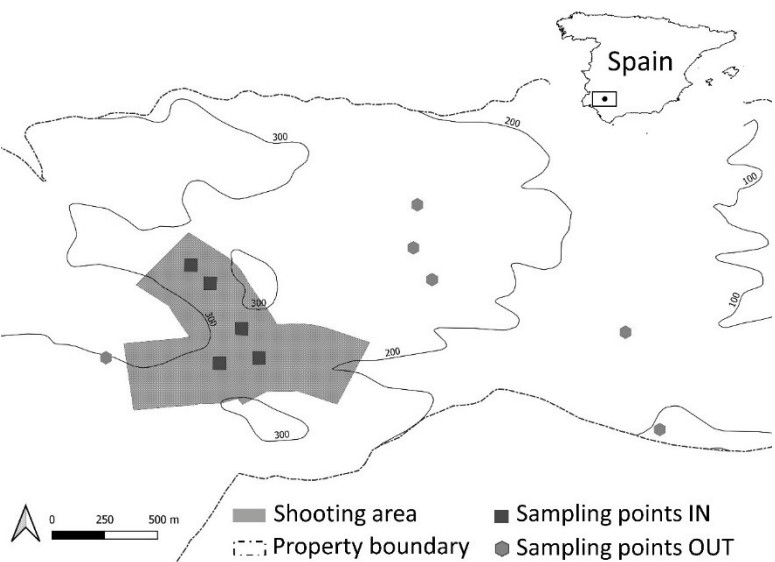

**Figure 1.** Location of the study area in Southern Spain, showing sampling vegetation sites. Black squares show the sampling points within the shooting area and grey hexagons show the sampling points outside the shooting area as controls.

The main economic activity in our study site is small-game hunting, concretely the red-legged partridge (*Alectoris rufa*). In the study area, the red-legged partridge is hunting in several modalities. The driven hunt modality called 'ojeo' is the main one. Other hunting species are the European rabbit (*Oryctolagus cuniculus*), common thrush (*Turdus philomelos*), mistle thrush (*Turdus viscivorus*) and red-rumped thrush (*Turdus pilaris*), wood pigeon (*Columba palumbus*), rock pigeon (*Columba livia*), collared dove (*Columba oenas*) and European turtle dove (*Streptopelia turtur*). As a secondary activity, big game hunting is mainly of red deer (*Cervus elaphus*) and wild boar (*Sus scrofa*).

The red-legged partridge driven hunts ('ojeo') are carried out on the dates set out by regional authorities for the practice of this special modality in hunting areas. The hunting season begins in early November and ends at the end of March. In the study area, approximately 36 driven hunts are carried out each year, spread over several days, each with between four and five positions or shooting lines. The shooting lines are usually made up

of eight hunters. Thirty people oversee beating the mountain to bring the red-legged partridges to the stands. Approximately 600 red-legged partridges are shot down each day, with an estimated 15,084 partridges per year. It means several shots, and with this, a large amount of Pb is dumping into the environment. The area where intensive red-legged partridge hunting is practiced is 158.5 ha (see Shooting Area in Figure 1). The Pb discharged amount was estimated using a conservative criterion that considered three shots per individual red-legged partridge retrieved [49]. The average weight of each ammunition cartridge is 32 g and contains 250 pellets. The estimated amount of Pb discharged into the environment in the study area was 71,375.40 ammunition pellets, weighing 9.4 kg of Pb, per hectare per year.

### 2.2. Sample Collection

Vegetation samples were obtained from shooting (five points) and non-shooting (six points) sampling sites of the estate between 2017 and 2019 (18th June and 18th July). Samples were collected from 32 plants from herbaceous, shrub and tree species randomly selected. A total of thirteen plants were from five points inside the shooting area and seventeen from six points out of the shooting area. Old and young sprouts were collected to determine whether the Pb pellets used during partridge hunting have been accumulating in plants over the years. We also considered the species (see Table 1) to examine if the rate of total Pb concentration differs between species.

**Table 1.** List of plant species collected for the analysis of Pb concentration accumulated in young and old sprouts from the leaves and sample size over the years (2017–2019) where samples were taken for this study. *n* refers to the total number of plants sampled each year (with young and old sprouts collected from each of them).

| Species | Description | 2017 | | 2018 | | 2019 | |
|---|---|---|---|---|---|---|---|
| | | *n* = 8 | | *n* = 10 | | *n* = 11 | |
| | | Shooting Area | Non-Shooting Area | Shooting Area | Non-Shooting Area | Shooting Area | Non-Shooting Area |
| *Cistus salviifolius* | Shrub/Perennial | 0 | 0 | 0 | 2 | 1 | 1 |
| *Cistus monspeliensis* | Shrub/Perennial | 1 | 1 | 0 | 0 | 0 | 0 |
| *Nerium oleander* | Shrub/Perennial | 0 | 0 | 1 | 0 | 0 | 0 |
| *Olea europaea* | Tree/Perennial | 0 | 2 | 1 | 1 | 1 | 2 |
| *Phillyrea angustifolia* | Shrub/Perennial | 0 | 0 | 1 | 0 | 0 | 0 |
| *Phlomis purpurea* | Shrub/Perennial | 0 | 0 | 1 | 0 | 0 | 0 |
| *Pistacia lentiscus* | Shrub/Perennial | 0 | 1 | 0 | 0 | 1 | 1 |
| *Quercus ilex* | Tree/Perennial | 1 | 1 | 1 | 1 | 1 | 1 |
| *Rubus ulmifolius* | Shrub/Perennial | 1 | 0 | 0 | 1 | 1 | 1 |
| Total sampled plants | | 3 | 5 | 5 | 5 | 5 | 6 |

### 2.3. Chemical Analyses

In the laboratory, the specimens were washed several times with bidistilled water to remove the remaining soil particles adhered to their surface. The determination of Pb concentration in leaves was achieved by atomic absorption spectrometry. The sample was milled and dried and then subjected to acid digestion using nitric acid and hydrogen peroxide, at a temperature of 200 °C (Ethos Plus microwave—Milestone, from Milestone Srl, Sorisole (BG)—Italy). Once the sample had been digested, its Pb content was determined in an AnalytiK Jena Atomic Absorption Spectrometer mod. ContrAA 700, equipped with a graphite chamber (GF). A calibration curve was developed with 5 Pb standards ranging from 2 ppb to 10 ppb. Pd chloride and Mg nitrate were used as a matrix modifier during the analysis to optimize the analytical conditions for better response in the GFAA

instrument and to lower the detection limits as well as to minimize interferences. The concentration of Pb was calculated as mg/Kg.

*2.4. Statistical Analyses*

All data were analyzed using R v. 3.6.1 (R Foundation for Statistical Computing, Vienna, Austria). Count variable "Pb concentration" was log-transformed (natural logarithm transformation) to improve the normality of residuals and to reduce skewness.

First, we ran a linear mixed model (LMM1), where we tested for potential differences in Pb concentration between both sampled areas (non-shoot vs. shoot) and sprout plants (young vs. old) using Pb levels log-transformed as a response variable. The model also included "individual nested into species", "species", and "year" as random terms to control repeated sampling of the same species in different years that vegetation samples were collected. A second linear mixed model (LMM2) was carried out to explore the effect of sampled species on Pb levels. For this model, a subset of data was used to use a significant sample of all species in both the shoot and non-shoot as control areas. With this aim, data from species sampled in both zones and repeated in more than one year were used (but see Table 1). This subset excluded data from 2017, and just includes four species: *O. europaea*, *R. ulmifolius*, *Q. ilex* and *C. salviifolius*. Pb levels log-transformed was used as a dependent variable and these four species as a factor controlling for both the area (non-shoot vs. shoot) and the sprout (young vs. old). It also included both interactions: sprout and species, to examine if the amount of Pb differential among sprouts is not equal for all species; and zone and species, to determine if the differences in Pb levels between areas are consistent across species.

Models were performed using the "lme4" package [50] and, in both case,s we removed non-significant interactions sequentially (*p*-value > 0.05) following a backwards-stepwise selection procedure in both models to avoid risks of over-parameterization. Models' results were presented for all main effects and significant interactions. In the figures we present mean predicted values from the LMM1 and LMM2 respectively. The means are given ± SE and the level of statistical significance was *p* < 0.05. Predictions were visualized with 'ggeffects' [51] and 'ggplot2' (v. 3.1.1) was used for graphics [52].

**3. Results**

Table 2 summarizes the mean (±S.E.) concentration of Pb in the leaves of the nine different sampled species in shooting sites and in the control sites. The mean concentrations of Pb in *C. salviifolius* and *O. europaea* showed significant differences between both locations. In all cases, the concentration of Pb was higher in individuals from shooting sites than from non-shooting sites. No differences between both sites were detected in Pb concentrations of *Q. ilex*, *C. monspeliensis*, *P. lentiscus* and *R. ulmifolius*.

**Table 2.** Mean ± S.E. Pb concentration (mg/Kg) in leaves of plant individuals classified by species and sample zones (shooting vs. non-shooting areas) during three consecutive years (2017–2019). Significant effects (*p*-value < 0.05) are shown in bold.

| Species | Mean (±S.E.) Pb Concentration (mg/Kg) | | *F*-Value | *p*-Value |
|---|---|---|---|---|
| | Shooting Area | Non-Shooting | | |
| *Cistus salviifolius* | 8.48 ± 2.81 | 1.06 ± 0.14 | $F_{1,8} = 41,64$ | **<0.001** |
| *Cistus monspeliensis* | 0.28 ± 0.03 | 0.38 ± 0.08 | $F_{1,2} = 1.37$ | 0.362 |
| *Nerium oleander* | 0.306 ± 0.071 | No sample | | |
| *Olea europaea* | 1.23 ± 0.825 | 0.169 ± 0.018 | $F_{1,12} = 4.737$ | **0.049** |
| *Phillyrea angustifolia* | 0.506 ± 0.010 | No sample | | |
| *Phlomis purpurea* | 0.677 ± 0.637 | No sample | | |
| *Pistacia lentiscus* | 0.305 ± 0.145 | 0.208 ± 0.092 | $F_{1,4} = 0.349$ | 0.587 |
| *Quercus ilex* | 0.746 ± 0.245 | 0.358 ± 0.053 | $F_{1,10} = 2.393$ | 0.153 |
| *Rubus ulmifolius* | 0.467 ± 0.186 | 0.601 ± 0.250 | $F_{1,6} = 0.183$ | 0.684 |

Results from the LMM1 are shown in Table 3. We found a significant effect of both sampling area and sprout of plants in the concentration of Pb. Vegetation of the shooting range showed higher Pb concentration than plants from control sites (Figure 2). In addition, we found significantly more Pb concentration in the old sprouts than in the young plants' sprouts (Figure 3).

**Table 3.** Results of LMM1 for the response variable Pb log- transformed concentration, and the effect of the sampled area (non-shooting area vs. shooting area area) and sprout of plants (young vs. old) considering the three sampled years (2017 to 2019). Reference levels for factors are shown in brackets. Significant effects (*p*-value < 0.05) are shown in bold.

| | Estimate | S.E. | d.f. | *t*-Value | *p*-Value |
|---|---|---|---|---|---|
| *Fixed factors* | | | | | |
| Intercept | −0.745 | 0.177 | 4.281 | −4.281 | **0.008** |
| Area (non-shooting area) | 0.227 | 0.093 | 2.447 | 2.447 | **0.022** |
| Sprout (young) | 0.313 | 0.077 | 4.092 | 4.092 | **<0.001** |
| *Random factors* | | | | | |
| Individual x Species: variance ± SE = 0.012 ± 0.110; Species: 0.072 ± 0.269; Year = 0.049 ± 0.222; Residual = 0.083 ± 0.288 | | | | | |

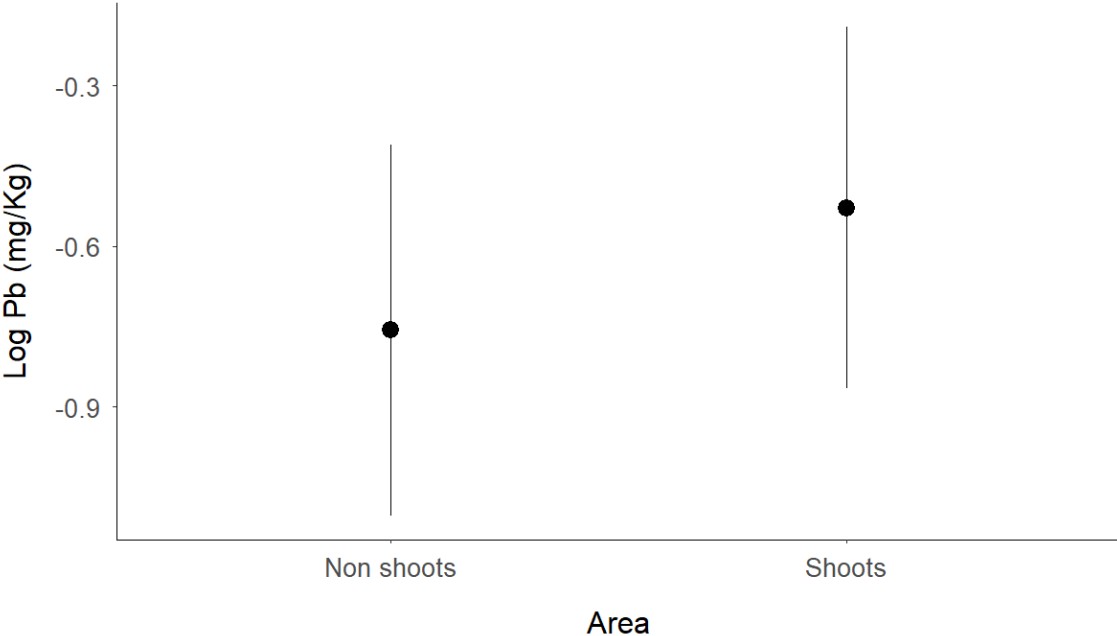

**Figure 2.** Pb concentration (mg/Kg) log-transformed in sixty-four plants of eleven different species in relation to the sampled area (non-shooting area vs. shooting area) in three sampled years. Pb log-transformed values refer to predicted values from LMM1 (Table 3). Data are shown as mean ± standard error.

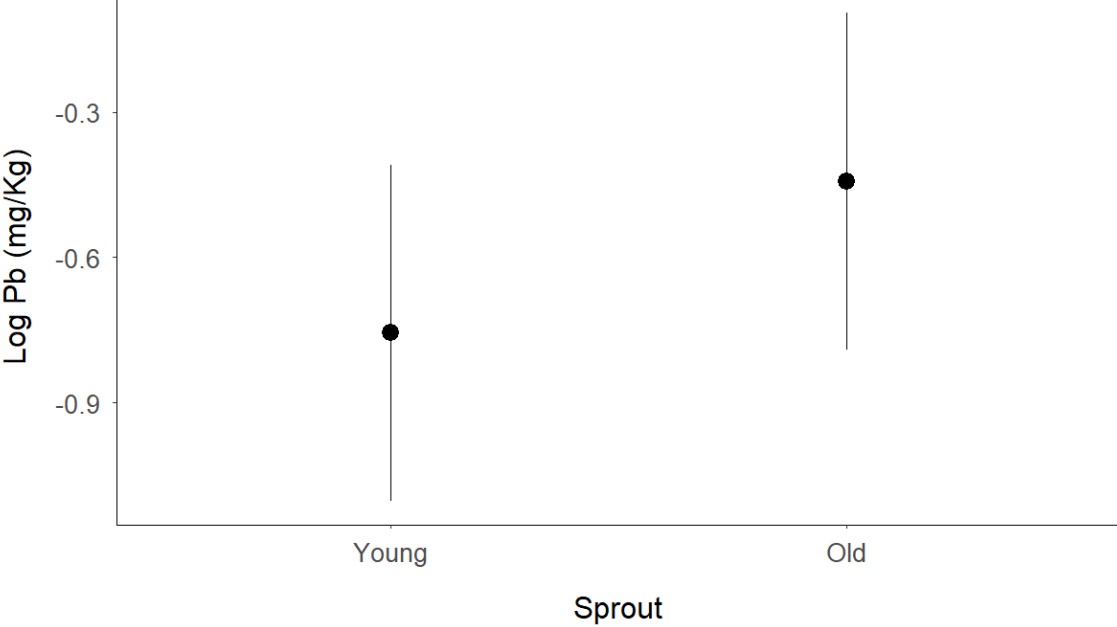

**Figure 3.** Pb concentration (mg/Kg) log-transformed in sixty-four plants of eleven different species in relation to the sprout of plants (young vs. old) in three sampled years. Pb log-transformed values refer to predicted values from LMM1 (Table 1). Data are shown as mean ± standard error.

Derived from the LMM2 (Table 4), the sampling site, plants' sprouts, and species showed a significant effect in the Pb concentration. According to LMM1 results, this model showed that shooting range plants in the shooting area and the old plants' sprouts have higher Pb concentrations. In addition, we found a significant effect of the sampled species on the Pb concentrations. In Figure 4 it is shown that *C. salviifolius* exhibits a higher Pb concentration of Pb than the other species independently of the sampled site (non-shooting vs. shooting) and plants' sprouts (young vs. old).

**Table 4.** Results of LMM2 for the response variable Pb concentration (mg/Kg) log-transformed, and the effect of the species and the interaction between sprout and species controlled by the sampled area (non-shooting area vs. shooting area) and sprout of plants (young vs. old) regarding the two sampled years (2018 to 2019). reference levels for factors are shown in brackets. Significant effects (*p*-value < 0.05) are shown in bold.

|  | Estimate | S.E. | d.f. | *t*-Value | *p*-Value |
|---|---|---|---|---|---|
| *Fixed factors* | | | | | |
| Intercept | −0.243 | 0.171 | 1.031 | −1.418 | 0.385 |
| Area (non-shooting area) | −0.205 | 0.069 | 11.062 | −2.964 | 0.013 |
| Sprout (young) | −0.158 | 0.037 | 16.000 | −4.243 | < 0.001 |
| Species | −0.267 | 0.106 | 11.000 | −2.527 | 0.004 |
| *Random factors:* | | | | | |
| Individual: variance ± SE = 0.045 ± 0.213; Year = 0.049 ± 0.222; Residual = 0.047 ± 0.217 | | | | | |

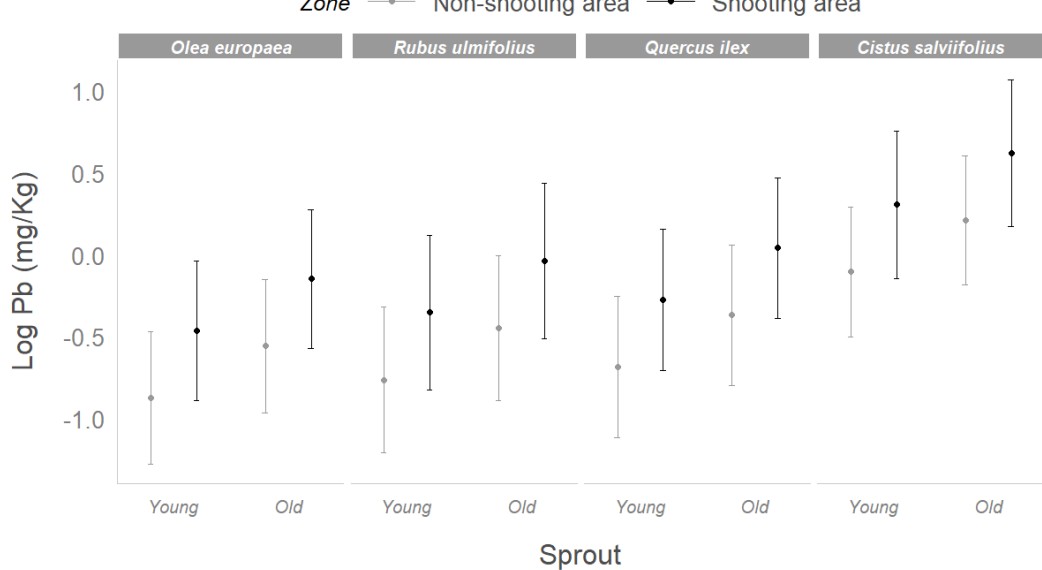

**Figure 4.** Pb concentration (mg/Kg) log-transformed in sixteen plants in relation to different species within both sprout of plants (young vs. old) in the two sampled years (2018–2019). Pb log-transformed values refer to predicted values from LMM2 (Table 4). Data are shown as mean ± standard error.

## 4. Discussion

This study shows the effect of Pb contamination from hunting ammunition on the vegetation from the red ledged partridge shooting range. We demonstrated the Pb accumulation over time due to the differences found between young and old sprouts in the same plant individuals. However, the effect of Pb ammunition in bird hunting areas is not equal depending on the sampled species. Despite the same exposure to this pollutant, this result suggests that species tend to sorb or accumulate Pb differently in their leaves.

Previous studies have revealed the accumulation of metals and metalloids in different animal tissues of numerous animal species [53–56]. In particular, some studies have already shown the adverse effect at different levels that Pb pellets derived from hunting actions have on the populations of several species of waterbirds ([32,57–59]; *Anas platyrhynchos* [60,61]), birds of prey (*Milvus milvus, Aquila adalberti* [62]; *Haliaeetus albicilla* [63]; *Aquila chrysaetos* [64]; *Buteo buteo* [65]), other bird species ([66,67] *Columbia livia* [68,69]; *Alectoris chukar* [70–72]; *Alectoris rufa* [49,73–75]; *Perdix perdix* [76]; *Phasianus colchicus* [77,78], and also mammals (*Lynx pardinus, Herpestes ichneumon, Meles meles, Genetta genetta* [79]; *Sus Scrofa* [80]; *Capra pyrenaica* [81]).

However, the study of the impact of Pb and other heavy metal contamination on vegetation from intensive hunting ranges are the cornerstones for the control and management of Pb pollution in shooting areas [18,82,83]. This work found that Pb plant concentrations were significantly higher in red-legged partridge shooting areas than in control sites. Consistent with these results, previous studies evidenced high Pb vegetation accumulation in mining areas [36,84], possibly due to aerial deposition from a large amount of Pb-containing dust in these sites [85–87]. In contrast, we speculate that the Pb accumulation we observed in plants' sprouts is due to Pb uptake from Pb deposition by pellets to the soil and translocation from the roots to the leaves [88,89]. It is to be expected that the uptake process is not immediate from the time of the hunting action. Generally, the accumulation of toxic heavy metals from the roots to the aerial parts of the plant, and especially to the reproductive structures, is restricted by several mechanisms (e.g., ability to reduce metal translocation from roots to shoots) [90–93].

We found that Pb concentration in old sprouts is higher than in young sprouts on same individuals, which supports the hypothesis that Pb accumulates over time in plants. That point is particularly relevant at sites where hunting is not sporadic but occurs intensively over the years. In the last few years in our study site, an average of ca. 15,082 red-legged partridges per year were caught, i.e., a very high number of red-legged partridges shot. Thus, the hunting pressure is high while the area where bird hunting is practiced is quite reduced (approximately 158.5 ha.). This implies a high concentration in the number of shots and, thus, in the amount of Pb spillage by pellets into this environment.

Future studies could examine the amount of Pb absorbed by plants from Pb in both plant roots and soil and vegetation affected by Pb deposition from Pb pellets. It must also investigate the role of other elements such as the presence of water points in the dispersion of Pb in the environment in specific shooting ranges. The work also throws light on further research using new approaches, such as Pb isotopes [19,21]. Pb ammunition is composed of several Pb sources (mainly recycled Pb) and may pose problems for the Pb footprint identification and the Pb source. However, using Pb isotope analysis could improve these results as it avoids other unstudied issues such as atmospheric dust supporting differences between the studied sites.

In this work, the species with the highest Pb concentrations in both young and old shoots were *Q. ilex* and *C. ladanifer*, in concordance with previous results carried out by Reglero and colleagues, 2018 [36]. Similarly, *O. europaea* and *R. ulmifolius* were the species with the lowest Pb concentrations in leaves. Previous works also found low Pb concentrations in *R. ulmifolius* at sites contaminated by mining [36,94] and other human practices [95]. This result is highly relevant for several reasons. The first reason lies in the importance of the use of native plants in phytoremediation processes in areas previously contaminated by human practices, with intensive hunting being one of them. It is decisive to find native plants for this purpose because these plants are often better in terms of survival, growth, and reproduction under environmental stress than plants introduced from another environment [95]. Secondly, these results allow us to study the impact of hunting on the sampled vegetation and other organisms indirectly affected by ingesting contaminated plants. Wild ungulates or domestic livestock may be affected by ingesting Pb from browsed plants during the period of food and pasture shortage. Species such as *C. ladanifer* and *Q. ilex* are browsed by red deer for more than three months in our latitudes due to the scarcity of available pasture due to summer drought [96]. However, the Pb concentrations in species found in our study area did not appear to be within the phytotoxicity range (30–300 mg/Kg) according to Pugh et al. (2002) [97] and, therefore, we cannot assure that this is a potentially toxic concentration for wild or domestic livestock. In addition to direct toxicity, Pb can interfere with the metabolism of certain essential elements including copper, zinc, and selenium by affecting their absorption, distribution, and bioavailability [98]. A more detailed study on Pb accumulation in different grass species (i.e., *Gramineae*) and other woody species commonly browsed by ungulates should be analyzed to examine the

indirect effect of Pb in these species and its toxicity derived from intensive hunting in our study area.

Our study has some relevant management implications since the hunting estate where intensive hunting of red-legged partridge takes place lies in an area of high ecological value and with ecosystems in good conservation status. This model requires regular analysis of the possible impacts on the environmental values of the zone, so that the management can guarantee compatibility between economic exploitation, the generation of wealth for the area of influence, and the conservation of environmental quality and biodiversity, including the populations of existing game species in the area. This type of management is not compatible with preserving natural populations of species such as the red-legged partridge. Thus, it must be contemplated when making decisions on the conservation of this activity in zones where there are native populations of red-legged partridge in the wild.

## 5. Conclusions

This work showed the impact of Pb ammunition from intensive hunting activity on vegetation regarding the persistence over time and the differential effect between species. These results will shed light on future studies that will examine the chemical and ecotoxicological impact of ammunition Pb on plants and the indirect impact of ingestion on animals in areas where intensive hunting is conducted. Our results provide new support in favor of the use of alternative ammunition, especially in areas where hunting activity is intensive.

**Author Contributions:** Data curation, formal analysis, investigation, writing—original draft, writing—review & editing: E.d.l.P.; methodology: E.d.l.P., J.M.S. and J.C.; funding acquisition, investigation, resources, supervision, validation: J.C.; conceptualization: E.d.l.P. and J.C. All authors have read and agreed to the published version of the manuscript.

**Funding:** This research was funded by UIRCP own research funds from the University of Córdoba.

**Institutional Review Board Statement:** Not applicable.

**Informed Consent Statement:** Not applicable.

**Data Availability Statement:** Not applicable.

**Acknowledgments** We thank the autonomous government of Andalucía (Junta de Andalucía) and the property of the hunting estate for providing funding and facilities to carry out this work. We thank Ángel Aparicio and Azahara Gómez for helping in sample collection.

**Conflicts of Interest:** The authors declare no conflict of interest.

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
