# Peer review of "The Impact of Pb from Ammunition on the Vegetation of a Bird Shooting Range"

_sustainability, doi:10.3390/su14053124_

Round 1
Reviewer 1 Report
Manuscript ID: Sustainability-1594483
Title- The impact of Pb from ammunition on the vegetation of bird shooting range
- Abstract should be clearer and in concise manner with more emphasizing the quantitative information.
- Recheck the keywords and try to follow the journal guideline and correct it.
- Introduction part provides only general information. It should be more updated with some recent references.
- There are a lot of review papers published in this proposed area, what is the difference between this manuscript and the previous published research papers by other researchers?
- Conclusions should highlight the insights and the applicability of your findings/results for further work. The author's own opinions too little summary about future prospects points of research in the manuscript.
- It would be better updating with some recent references with 2020-21 latest publication and avoid the old references.
- Several typesetting errors are present throughout the text and stylistic features should be homogenized as well.
- Manuscript formatting, references, abbreviation, does not proper format, carefully re-visit the journal guideline.
https://www.sciencedirect.com/science/article/abs/pii/S0048969718339202
https://www.sciencedirect.com/science/article/abs/pii/S0013935119305894
https://www.mdpi.com/1660-4601/17/7/2179
Author Response
Reviewer #1
- Abstract should be clearer and in concise manner with more emphasizing the quantitative information. >>> We agreed with your comment, so we have modified the entire abstract to enhance the obtained results. We hope that these changes have improved the abstract, but we are willing to make any needed changes in future revisions.
“Hunting with lead ammunition represents a source of heavy metal pollution to the environment that can be potentially high at the local scale. Intensive hunting on small game species can con-centrate high levels of ammunition discharging on small areas. This type of hunting is a relevant economic resource for private landowners in some regions of Spain, and current legislation allows the use of lead ammunition in these scenarios. It becomes, therefore, highly relevant to study whether this activity may pose concerns to the conservation of the environment in the areas where it takes place. Using a red-legged partridge (Alectoris rufa) shooting range as a study area, we examined the effect of intensive hunting on this species on the vegetation present. We found significantly higher lead levels in the sprouts of plants of shooting areas related to control sites of the same property where partridge shooting does not occur. We found differences in the presence of lead between sprouts of different plant species. In addition, old sprouts existing vegetation in shooting areas also showed higher lead levels than newly emerged sprouts of the same plants. These results demonstrate the impact of lead ammunition on vegetation in terms of persistence over time and differences between species. Further analyses using chemical and ecotoxicological data are necessary to evaluate the extent of environmental pollution risks. Our results provide new support in favour of the use of alternative ammunition, with particular emphasis on scenarios where hunting activity is intensive.”
- Recheck the keywords and try to follow the journal guideline and correct it. >>> OK, done.
- Introduction part provides only general information. It should be more updated with some recent references. >>> OK, we have updated this part with recent references.
- There are a lot of review papers published in this proposed area, what is the difference between this manuscript and the previous published research papers by other researchers? >>> We agree with you. However, our paper shows a different scenario than previous articles have done: the intensive hunting activity of red-legged partridge in a reduced area. We have added a paragraph in the introduction where we explained the relevance of our paper compared to other works.
- 97-107: “In Spain, intensive hunting is restricted to specific sites. The main purpose is hunting using periodic releases of small game reared on certified hunting farms, where appropriate, or in which small game species are regularly restocked. This context requires the analysis of the interactions between the management model and the effects on the environmental values surrounding these areas. The setting of a management protocol is needed to ensure compatibility between an economic benefit and the maintenance of the environment and biodiversity conservation. One of the most relevant impacts in these scenarios is the discharge of elevated ammunition due to the concentration of intensive hunting activity in time and space. Thus, investigating the short- to medium-term effects on the environment (e.g. vegetation) is key to monitoring and evaluating its impact.”
- Conclusions should highlight the insights and the applicability of your findings/results for further work. The author's own opinions too little summary about prospects points of research in the manuscript. >>> OK, we have added conclusions epigraph.
- 366-373: “This work showed the impact of Pb ammunition from intensive hunting activity on vegetation regarding the persistence over time and the differential effect between species. These results shed light on future studies to examine the chemical and ecotoxicological impact of ammunition Pb on plants and the indirect impact of ingestion on animals in areas where intensive hunting is conducted. Our results provide new support in favour of the use of alternative ammunition, especially in areas where hunting activity is intensive.”
- It would be better updating with some recent references with 2020-21 latest publication and avoid the old references. >>> We agree with you. We have added some recent references.
- Several typesetting errors are present throughout the text and stylistic features should be homogenized as well. >>> OK, changed.
- Manuscript formatting, references, abbreviation, does not proper format, carefully re-visit the journal guideline. >>> OK, revisited.
https://www.sciencedirect.com/science/article/abs/pii/S0048969718339202
https://www.sciencedirect.com/science/article/abs/pii/S0013935119305894
https://www.mdpi.com/1660-4601/17/7/2179
(x) Extensive editing of English language and style required >>> Thank for your comments and suggestions. We send the manuscript to a native translator to improve the language. We hope to have improved the quality of the manuscript, but we are prepared to make any necessary changes.

Reviewer 2 Report
In my opinion it is the interesting manuscript. The knowledge about impact of hunting on natural environment is actual subject. Furthermore, heavy metals, especially Pb are important issues. However, the authors did not avoid mistakes.
- Abstract should be modified, Authors are asked to presents some quantitative values in order to enhanced the obtained results.
- Table 1. The lack description: Why were the other species not researched for the 2017-2019? The only one specie (Quercus ilex) was researched from shooting area and not-shooting area in the years 2017, 2018 and 2019.
- line 188-190: This is sentence from the author's guideline material.
- line 272-273: The lack description of “.. several mechanism.”
- There should be a part - conclusion. The conclusion should summarize the key deliverables.
- Authors should use latest references.
Author Response
In my opinion it is the interesting manuscript. The knowledge about impact of hunting on natural environment is actual subject. Furthermore, heavy metals, especially Pb are important issues. However, the authors did not avoid mistakes.
Abstract should be modified; Authors are asked to present some quantitative values in order to enhance the obtained results. >>> We agreed with your comment, so we have modified the entire abstract to enhance the obtained results. We hope that these changes have improved the abstract, but we are willing to make any needed changes in future revisions.
“Hunting with lead ammunition represents a source of heavy metal pollution to the environment that can be potentially high at the local scale. Intensive hunting on small game species can con-centrate high levels of ammunition discharging on small areas. This type of hunting is a relevant economic resource for private landowners in some regions of Spain, and current legislation allows the use of lead ammunition in these scenarios. It becomes, therefore, highly relevant to study whether this activity may pose concerns to the conservation of the environment in the areas where it takes place. Using a red-legged partridge (Alectoris rufa) shooting range as a study area, we examined the effect of intensive hunting on this species on the vegetation present. We found significantly higher lead levels in the sprouts of plants of shooting areas related to control sites of the same property where partridge shooting does not occur. We found differences in the presence of lead between sprouts of different plant species. In addition, old sprouts existing vegetation in shooting areas also showed higher lead levels than newly emerged sprouts of the same plants. These results demonstrate the impact of lead ammunition on vegetation in terms of persistence over time and differences between species. Further analyses using chemical and ecotoxicological data are necessary to evaluate the extent of environmental pollution risks. Our results provide new support in favour of the use of alternative ammunition, with particular emphasis on scenarios where hunting activity is intensive.”
Table 1. The lack description: Why were the other species not researched for the 2017-2019? The only one specie (Quercus ilex) was researched from shooting area and not-shooting area in the years 2017, 2018 and 2019. >>> The plants included in this study were chosen randomly in each sampling (L.166: “Samples were collected from thirty-two plants from herbaceous, shrub and tree species randomly selected”). This is the reason that not all species are represented in all sampled years. We feel that it would be optimal to have sampled different individuals of all species for each year studied. Future work should address this improvement to support inter-species differences in ammunition Pb concentration in the vegetation.
line 188-190: This is sentence from the author's guideline material. >>> OK, thank you. We have already deleted this sentence. Sorry about the mistake.
line 272-273: The lack description of “.. several mechanism.” >>> OK, added: L.305 “several mechanisms (e.g., ability to reduce metal translocation from roots to shoots)”.
There should be a part - conclusion. The conclusion should summarize the key deliverables. >>> OK, added. Thanks. L. 363-373: “”.
Authors should use latest references. >>> We agree with you. We added some latest references about the study topic.

Reviewer 3 Report
This paper is excellent. However, the use of "we" in most sentences is not suitable. Kindly rewrite these sentences.
Line 151: "Nitric acid and hydrogen peroxide, at a temperature of 200 ºC, were used to digest the sample." Kindly include equipment used with brand and model to heat the sample until 200 ⁰C.
Author Response
Reviewer #3
This paper is excellent. However, the use of "we" in most sentences is not suitable. Kindly rewrite these sentences. >>> Thanks for your comments. We have rewritten most part of the manuscript regarding this and other issues. We have made all the changes that you and the other reviewers suggested. We hope it has improved but we are prepared to make the necessary changes. We also have used a professional translator that revise and improve the English of the manuscript.
Line 151: "Nitric acid and hydrogen peroxide, at a temperature of 200 ºC, were used to digest the sample." Kindly include equipment used with brand and model to heat the sample until 200 ⁰C. >>> We included this information in the previous sentences: “The sample was milled and dried and then subjected to an acid digestion in an Ethos Plus microwave digester (Milestone).”
However, we have changed this paragraph to ensure that this information is clear to the readers. L. 183-186: “The sample was milled and dried and then subjected to acid digestion using nitric acid and hydrogen peroxide, at a temperature of 200 ºC (Ethos Plus microwave - Milestone).”

Round 2
Reviewer 1 Report
The manuscript is significantly improved, most comments are properly addressed.